# Analyzing and predicting global happiness index via integrated multilayer clustering and machine learning models

**Boxu Yang***, **Xiang Xie**

School of Economics and Management, Beijing Jiaotong University, Beijing, China

* 24120587@bjtu.edu.cn

## Abstract

This study addresses the research objective of predicting global happiness and identifying its key drivers. We propose a novel predictive framework that integrates unsupervised and supervised machine learning techniques to uncover the complex patterns underlying happiness scores across nations. Initially, we apply K-Means clustering to group countries based on similarities in their happiness patterns. For the first time, these cluster assignments are subsequently incorporated as additional features into ensemble learning models—specifically, Random Forests and XGBoost—to enhance the prediction of happiness scores. This hierarchical analysis approach yields a significant improvement in predictive performance, with an approximate 12% increase in R² compared to models that do not include clustering information. Using data from the World Happiness Report, our analysis reveals that global happiness can be categorized into three distinct groups (high, medium, and low). Among the various determinants examined, social support and GDP emerge as the most influential factors contributing to the happiness index. These findings not only advance the methodological framework for predicting happiness but also provide robust evidence for policymakers seeking to implement targeted interventions aimed at improving public well-being and promoting social progress.

## 1 Introduction

### 1.1 Research background

Happiness is a complex and subjective concept that involves the overall satisfaction of individuals and groups on both socio-economic and psychological levels. As a global research project, the World Happiness Report has provided extensive data since 2012, covering variables such as GDP, social support, healthy life expectancy, freedom, generosity, and perceptions of corruption. These indicators form the foundation for quantifying the happiness index [1]. The happiness index has gradually become a core measure of a country's socio-economic development, and its multi-dimensional characteristics make its analysis crucial for both academic research and policy-making, especially in modern societal development [2]. Richard Easterlin's (1974) study first revealed that economic growth is not the only driving factor for improving happiness levels, prompting scholars to further explore the social and economic determinants of happiness [3]. In recent years, multi-level studies on happiness data using

Kaggle platform. The datasets include: World Happiness Report Data: https://www.kaggle.com/datasets/unsdsn/world-happiness The datasets used in this study are derived from the World Happiness Report (2020-2024) and include data from 156 countries and regions, covering variables such as GDP per capita, social support, healthy life expectancy, freedom to make life choices, generosity, and perceptions of corruption. All relevant data are publicly available and can be accessed without restrictions, ensuring the reproducibility of the study. The processed and cleaned data, along with the code used for analysis, are available upon request to facilitate transparency and further research.

**Funding:** This work was supported by the National Natural Science Foundation of China "Joint Fund Project" (Research on Intelligent High-Speed Rail Data Service System Based on Data Rights Confirmation) under Grant U2268202 (Project Coding: B22A1500010). The funders had no role in study design, data collection and analysis, decision to publish, or preparation of the manuscript.

**Competing interests:** The authors have declared that no competing interests exist.

machine learning techniques and clustering analysis have provided new perspectives for understanding the intrinsic patterns of the happiness index. Although the happiness index holds a significant place in policy and social science research, there are still deficiencies in its analytical methods [4]. Traditional methods, such as linear regression, have limited research on the interactions between happiness variables, and the application of unsupervised learning and machine learning methods, such as clustering analysis, is still in the exploratory stage. For example, Chakraborty and Tsokos (2021) used K-Means clustering to uncover patterns in national happiness indices but did not delve into the specific causal relationships between variables [5]. The COVID-19 pandemic has had a significant impact on global happiness, particularly in terms of the dynamic changes in social support and mental health [6]. Research has shown that happiness data during the pandemic experienced abnormal fluctuations, highlighting the need for the study of the happiness index to dynamically adapt to emerging social challenges. As globalization accelerates and social inequality intensifies, cross-national analysis and prediction of the happiness index have become particularly important [7]. This study aims to combine clustering analysis with machine learning models to explore the intrinsic patterns of the happiness index through multi-level analysis, providing a scientific basis for policy-making.

## 1.2 Literature review

### 1.2.1 Application of clustering analysis in happiness index research.

Clustering analysis is an unsupervised learning technique that discovers the potential group characteristics in data and is widely used in social and economic research. This method can classify groups according to specific indicators, thereby revealing the characteristic differences between different categories. Rendón et al. (2011) pointed out that the validity indicators in clustering analysis, such as cohesion and separation, are important tools for evaluating the quality of clustering [8]. In addition, the NbClust tool developed by Charrad et al. (2014) can automatically select the number of clusters through multiple validity indices [9], providing strong support for the analysis of complex social data.

Although clustering analysis is widely used in the social and economic fields [10], there are relatively few in-depth analyses of the happiness index. Some studies have attempted to identify the patterns of the happiness index through K-Means clustering, but often only stayed at the descriptive classification level and did not deeply analyze the interaction of variables between groups. For example, Oswald et al. (2015) emphasized the correlation between happiness and productivity and suggested combining clustering results with economic and social data to explore the driving factors of happiness [11]. In addition, Freeman and Di Tella (2006) pointed out that clustering analysis should be combined with causal inference tools to further enhance its explanatory power [12].

### 1.2.2 Application of machine learning models in happiness index prediction.

Machine learning models, due to their strong predictive ability, are widely used in the analysis and prediction of social variables. The Random Forest algorithm proposed by Breiman (2001) has good interpretability and robustness and has been proven to be suitable for modeling nonlinear and high-dimensional data [13]. In addition, the XGBoost algorithm developed by Chen and Guestrin (2016) has high efficiency and robustness to data distribution and has become a popular tool in social and economic research [14].

In the research of the happiness index, machine learning technology has begun to emerge. For example, Howell (2008) revealed the positive correlation between economic status and subjective well-being through the regression tree model [15]. Other studies have shown that Random Forest and XGBoost perform well in predicting the happiness index and can quantify

the importance of variables, such as the core role of GDP and social support in happiness. However, most of the existing literature focuses on a single method and rarely integrates clustering analysis and machine learning models for multi-level comprehensive research.

Current research has revealed the complementarity of clustering analysis and machine learning models in social and economic research, but in the research of the happiness index, the integrated application of the two is still insufficient. Especially in the dynamic analysis of the happiness index, existing research mainly focuses on static data and ignores the time dimension and the complex interaction between variables [16]. Therefore, it is necessary to further combine clustering analysis and machine learning to explore the multidimensional characteristics and dynamic changes of the happiness index.

**1.2.3 Gaps and innovations.** Despite extensive research utilizing either clustering analysis or machine learning prediction methods for happiness index studies, most prior work has been limited to a single methodological approach. Some studies have employed clustering techniques, such as K-Means, solely for descriptive classification of national happiness characteristics, thereby revealing only superficial group differences. Conversely, other studies have independently applied machine learning models—such as Random Forests and XGBoost—to predict the happiness index and explore the relationships between economic and social variables and happiness. The limitation of using clustering methods in isolation lies in their inability to capture the complex interdependencies among variables, while standalone prediction models may suffer from a lack of higher-order structural information.

This study introduces, for the first time, a two-stage "clustering–prediction" framework designed to fully leverage the strengths of both approaches. Specifically, the first stage employs clustering analysis to perform multi-level segmentation of global happiness data, thereby revealing differences across countries or regions in terms of happiness levels, sustainable development, and cultural values. In the second stage, the clustering results are incorporated as new features into machine learning prediction models to construct a hierarchical prediction framework. Ablation experiments demonstrate that this framework significantly improves prediction accuracy, effectively uncovering the multidimensional dynamic characteristics of the happiness index and the complex interactions among variables.

This integrated approach not only compensates for the limitations of traditional single-method studies in static description and prediction, but also, by incorporating time series data, explores the dynamic evolution of the happiness index—addressing the current literature's deficiency in temporal analysis. In summary, the innovative contribution of this study lies in the development and validation of a hierarchical integrated framework that combines clustering analysis with machine learning prediction, thereby providing novel theoretical and empirical support for in-depth happiness index research and related policy formulation.

## 1.3 Research objectives

The happiness index, as an important indicator for measuring national social and economic development, provides a comprehensive evaluation of social progress and quality of life. Existing research has limitations in theory and method in revealing the national differences in the happiness index and the key variables affecting its changes. This study combines clustering analysis and machine learning models to explore the multidimensional characteristics of the happiness index. The specific objectives are as follows:

**Objective 1: Explore the national differences in the happiness index through K-Means clustering analysis.** Based on the multi-dimensional data of the World Happiness Report, the K-Means clustering method is used to classify the happiness indexes of different countries to reveal the significant characteristic differences in the happiness of countries. This study will

combine the data distribution characteristics and the best clustering number determination techniques (elbow method and silhouette coefficient, etc.) to achieve the effective classification of high-happiness, medium-happiness, and low-happiness country groups, thus laying the foundation for subsequent in-depth analysis [17].

**Objective 2: Construct a machine learning model to identify the key variables affecting the happiness index.** Using mainstream machine learning algorithms such as Random Forest and XGBoost, a happiness index prediction model is constructed to quantitatively evaluate the relative importance of the variables affecting the happiness index. Focus on the relative importance of variables such as GDP, social support, and healthy life expectancy, and comprehensively compare the prediction capabilities of different algorithms combined with model performance indicators (such as mean square error MSE and coefficient of determination $R^2$) [18].

**Objective 3: Propose accurate prediction and policy suggestions.** By combining the research results of clustering analysis and machine learning models, this study will summarize the main driving factors of the happiness index and put forward targeted policy suggestions, especially providing practical paths for low and medium-happiness countries to improve their happiness levels. This objective aims to achieve the combination of academic research and policy application and further promote the practical significance and value of happiness index research.

## 2 Data and methods

### 2.1 Data source and sample

The data of this study comes from the World Happiness Report (2020–2024), which is published by the United Nations Sustainable Development Solutions Network (SDSN) and covers the social and economic indicators of 156 countries and regions around the world [19]. The data in the report is based on the questionnaire results of the Gallup World Poll and provides a systematic framework for measuring the happiness index. The happiness index is based on the Cantril Ladder method and is obtained by evaluating the subjective scores of respondents on their current living conditions [20], with a score range from 0 (the lowest quality of life) to 10 (the highest quality of life).

The variables used in the study cover economic, social, health, governance, and other aspects, specifically including:

1. Economic production (GDP per capita): Per capita gross domestic product, reflecting the economic level.

2. Social support: Measures whether an individual can rely on others when in need of help.

3. Healthy life expectancy: Per capita healthy life expectancy calculated based on WHO data.

4. Freedom to make life choices: Reflects the degree of freedom of an individual in choosing the direction of life.

5. Generosity: Based on whether there has been a donation to a charity or helping others in the past month.

6. Perceptions of corruption: Measures the respondents' perception of the degree of corruption in the government and business environment.

7. Dystopia Residual: Used as a benchmark score to adjust the part of the actual data that is not explained by the above variables.

The data preprocessing includes the following steps:

1. Missing value handling: For a small amount of missing data, the multiple imputation method is used to reduce bias [21].

2. Standardization processing: Normalize the variables with different dimensions to ensure balance in subsequent analysis.

3. Time series arrangement: Since this study involves data from nearly five years, the data of each year is integrated to ensure the continuity and consistency of the analysis.

The study will explore the national differences in the happiness index through the systematic analysis of these variables and construct a prediction model to identify the key driving factors.

## 2.2 Research methods

**2.2.1 Clustering analysis.** To explore the national differences in the happiness index, this study uses the K-Means clustering method to conduct unsupervised learning analysis on the data of the World Happiness Report. Through clustering analysis, this study can reveal the distribution pattern of the happiness index and provide data stratification support for the subsequent prediction model. The specific steps are as follows:

1. Preprocessing and standardization: Normalize each variable to reduce the impact of variable scale differences on the clustering results.

2. Determination of the number of clusters: Use four methods to determine the best number of clusters, such as the elbow method. By analyzing the change trend of the total sum of squared errors under different numbers of clusters, select the optimal number of clusters [22].

3. Clustering execution: Use the K-Means algorithm to cluster the multi-dimensional happiness index data of 156 countries into high, medium, and low happiness groups.

**2.2.2 Prediction model.** Based on the Random Forest and XGBoost algorithms, a happiness index prediction model is constructed to evaluate the relative importance and prediction ability of each variable on the happiness index.

**2.2.3 Model validation.** To evaluate the model performance and verify its prediction ability, this study uses the following indicators [23]:

1. Mean square error (MSE): Measures the average of the squared differences between the predicted values and the actual values. The smaller the value, the better the model performance.

2. Coefficient of determination ($R^2$): Used to evaluate the goodness of fit of the model, with a range from 0 to 1. The closer to 1, the stronger the model's explanatory ability.

The definitions are as follows:

$$MAE = \frac{1}{m} \sum_{i=1}^{m} |y_i - y_i'|^2 \tag{1}$$

$$MSE = \frac{1}{m} \sum_{i=1}^{m} (y_i - y_i')^2 \tag{2}$$

$$RMSE = \sqrt{MSE} \tag{3}$$

$$R^2 = 1 - \frac{\sum_{i=1}^{m} (y_i - y_i')^2}{\sum_{i=1}^{m} (y_i - \bar{y})^2}$$

(4)

where $m$ represents the total number of elements in the test data, $y_i'$ is the predicted value, $y_i$ is the corresponding real value of the $i$-th sample, and $\bar{y}$ is the average of the actual observed values. $R^2$ is used to measure the fitting degree of the regression model to the observed data, and its value is between 0 and 1. When $R^2$ is close to 1, it indicates that the model has a good fitting effect on the data, that is, the model can well explain the change of the dependent variable; when $R^2$ is close to 0, it indicates that the model has a poor fitting effect on the data and the model's explanatory ability for the dependent variable is weak.

**2.1.4 Research framework process.** The process of the research framework of this paper is shown in Fig 1.

## 3 Statistical analysis

### 3.1 Data description

Based on the data of the World Happiness Report, the distribution of the happiness index and its related variables of 156 countries from 2020 to 2024 was analyzed. In most parts of Europe, especially in Nordic countries such as Norway, Sweden, and Finland, the happiness scores are relatively high. Canada and the United States in North America also have relatively high happiness indexes. The situation in Asia is more complex. Some countries, such as China, show a transitional state, while countries such as India show relatively low happiness scores. In Africa, the scores are more diverse, reflecting the large differences in the happiness index. Some countries in South America, such as Argentina, have high scores, while others have low scores. Australia and New Zealand in Oceania mainly have a relatively high level of happiness.

When the 156 countries are classified by continent, Fig 2 is obtained. It can be clearly seen from the bar chart that Europe has the highest score of 7.58. Oceania has the second-highest score of 7.27. The Americas has a score of 6.63. Asia has a score of 6.19. Africa has a relatively low score of 4.77.

When analyzing the data by year, a box plot (violin plot) of the happiness scores from 2020 to 2024 is obtained [24]. It can be seen from Fig 3 that the distribution of happiness scores in each year is roughly similar, mainly concentrated between 5 and 7. The central tendency of the happiness scores has increased after COVID-19, from 4.7 in 2020 to 5.8 in 2024.

As shown in the heat map in Fig 4, from the perspective of variable correlation, Social support, Score, GDP per capita, and Healthy life expectancy show a relatively strong positive correlation, with a deep red color and a correlation coefficient above 0.7, indicating that these factors are closely related to each other and may jointly affect the overall happiness or development level [25]. The variables were screened by thermal maps and variance expansion factor (VIF<5) to exclude multicollinearity. Generosity has a relatively weak correlation with other variables, with a lighter color. Freedom to make life choices and Perceptions of corruption also have a certain positive correlation with other variables, but the degree is not as high as the previous variables. In terms of systematic clustering distribution, Social support, Score, GDP per capita, and Healthy life expectancy are relatively close in the clustering tree, indicating that they have high similarity and relevance in the data structure and can be classified into one category. [26] Generosity, Freedom to make life choices, and Perceptions of corruption are relatively independent but also have a certain connection with the previous category. Overall, this heat map clearly shows the relationships and structures between the variables through color and clustering tree. Fig 4 shows that the most important variables affecting the happiness score are Social support and Healthy life expectancy, with the highest correlation

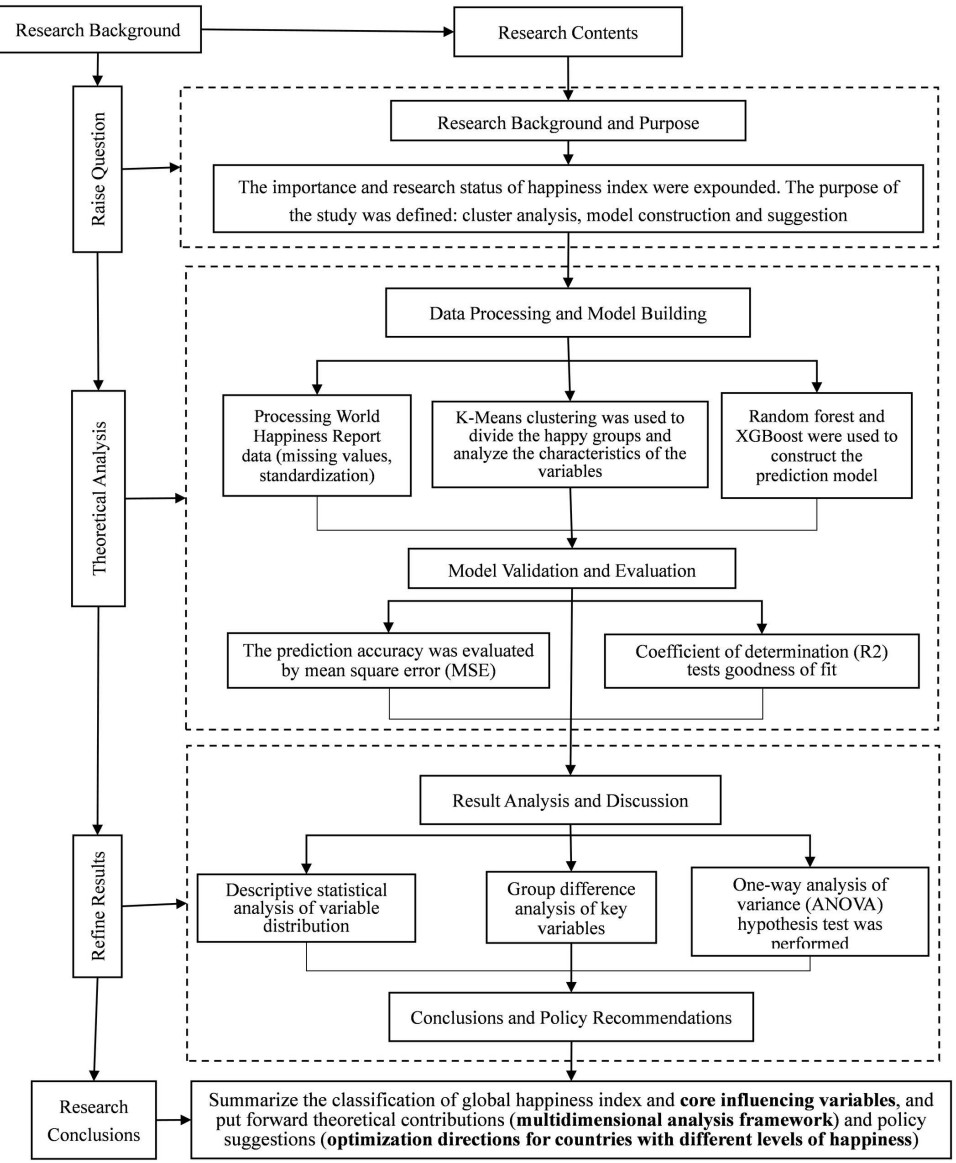

**Fig 1. Research Framework Process.**

coefficients with the happiness score of 0.79 and 0.78 respectively. The least important variable is Generosity, with a correlation coefficient with the happiness score of only 0.08. It provides key guidance for the selection of variables in machine learning to predict happiness scores.

## 3.2 Clustering analysis

Clustering analysis, as a common method in unsupervised learning, aims to automatically divide data into different groups through similarity measurement. In social science research, clustering analysis is widely used to explore the characteristic differences of different groups. For example, countries around the world can be grouped according to their happiness indexes. Through this method, researchers can identify countries with higher and lower

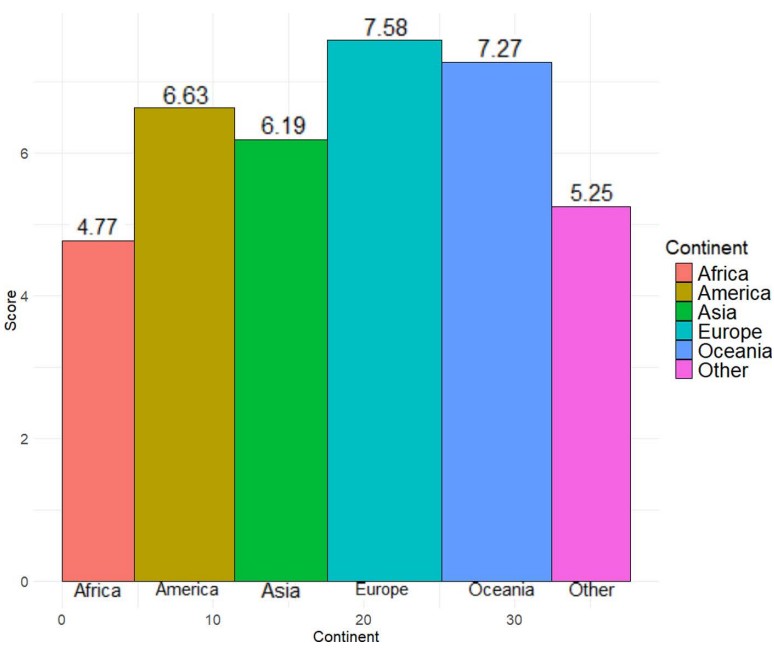

**Fig 2. Bar chart of average happiness scores across continents.**

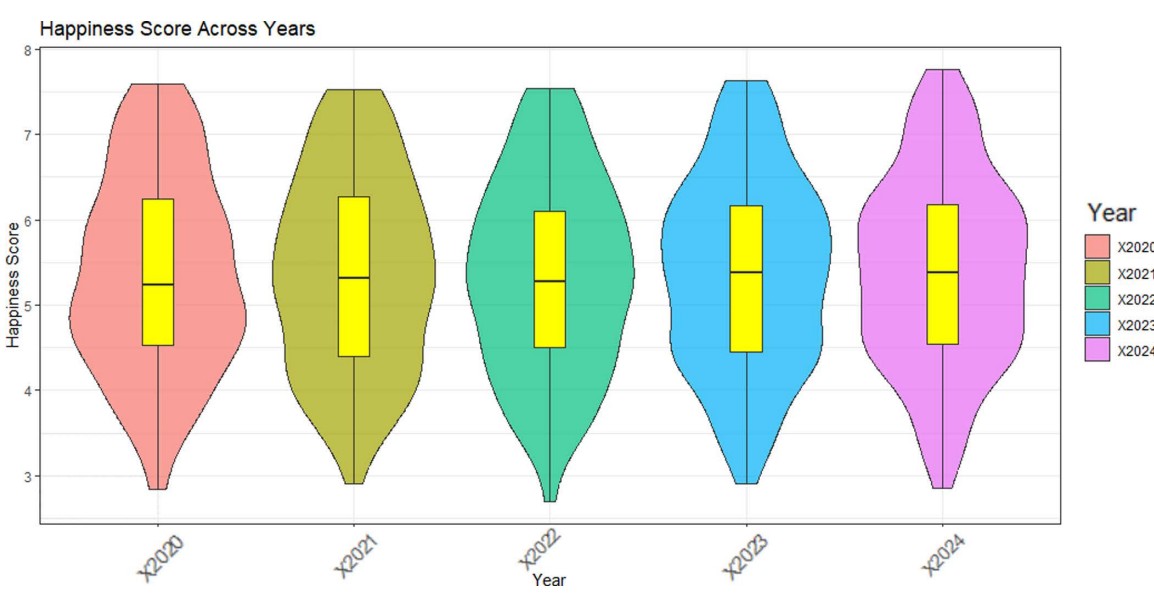

**Fig 3. Happiness score box graph for 2020-2024.**

happiness levels and their key characteristic differences in social, economic, and psychological aspects.

**3.2.1 Determination of the optimal number of clusters.** When conducting clustering analysis, it is first necessary to determine how many classes the data will be divided into, that is, the number of clusters. The K-Means algorithm is one of the most commonly used clustering methods. It mainly optimizes the clustering results by calculating the distance from

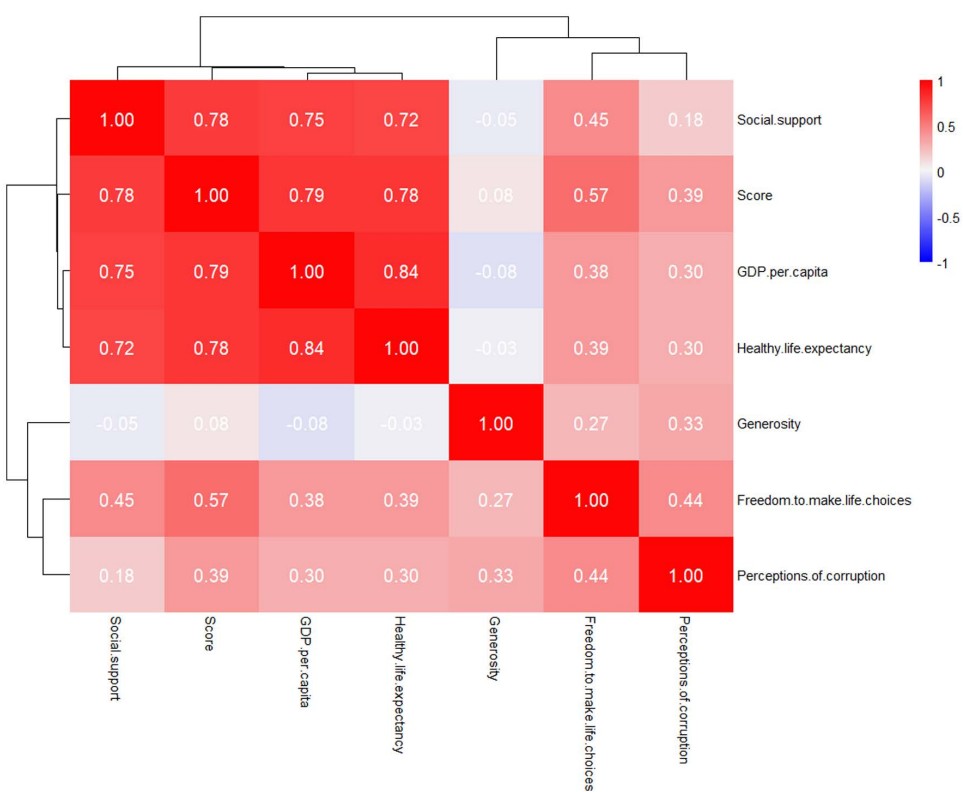

**Fig 4. Heat maps for each variable.**

the data points in the cluster to the centre. However, a key problem of the K-Means algorithm is that the number of clusters K needs to be specified in advance. To determine the optimal K value, the method of selecting the number of clusters is somewhat subjective and also depends on the technology used to calculate similarity and the parameters used for partitioning. [27] However, there are nearly 30 methods to determine the optimal number of clusters. This study will comprehensively use the most popular methods, including the elbow method, silhouette method, Hartigan and Gap statistics, to determine the optimal number of clusters [22]. As shown in Fig 5, the final optimal number of clusters is 3.

For example, the core idea of the elbow method is that as the number of clusters K increases, the total sum of squared errors of the clustering results will gradually decrease. However, when K reaches a certain threshold, increasing K will no longer significantly reduce the error [22]. By drawing the relationship graph between K and SSE, the "elbow" position, that is, the position where the rate of error reduction begins to slow down, can be observed and used as the selection criterion for the optimal number of clusters.

**3.2.2 Clustering analysis process.** Before conducting clustering analysis, we first standardized the data to eliminate the influence of dimensional differences between different variables. The standardized data ensures that the contribution of each variable to clustering is relatively balanced [28]. Based on the determined optimal number of clusters K = 3, we divided all countries into three categories: high happiness index group, medium happiness index group, and low happiness index group. The differences in happiness index within each group are relatively small, while significant differences are shown between different groups. The clustering result graph in Fig 6 projects the multi-dimensional data into three clusters

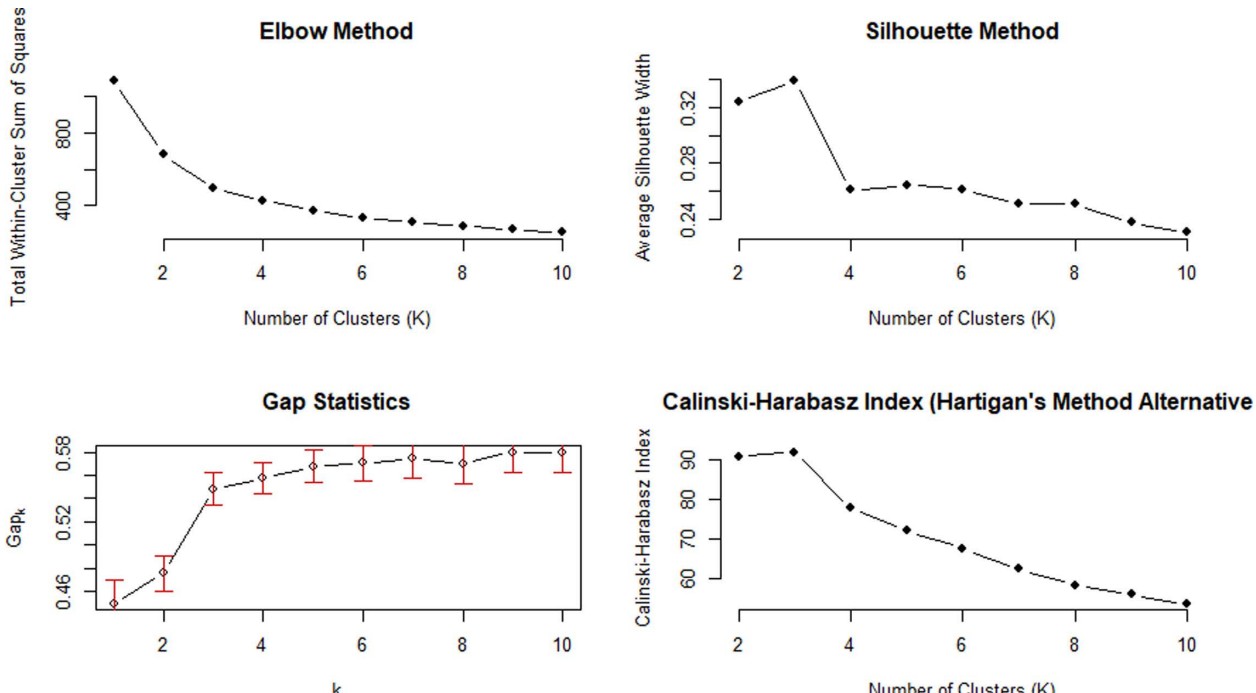

**Fig 5. Elbow Method (upper left); Average Silhouette Plot (upper right); Gap Statistics (lower left); Hartigan Plot (lower right) showing the optimal number of clusters.**

formed by k-means clustering, revealing significant differences in social and economic characteristics between different groups. For example, the high happiness index group usually has a high GDP, per capita healthy life expectancy, and social support level, while the low happiness index group often faces low economic development and social welfare guarantees.

**3.2.3 Key characteristic analysis between clustering groups and variables.** As shown in Table 1, when analyzing the three clusters, we can observe significant differences in their multiple key characteristics. First, the happiness score (Score) is an important indicator for distinguishing each cluster. The happiness score of Cluster 1 (high happiness group) is the highest, reaching 7.064, which is significantly higher than that of Cluster 2 (low happiness group) and Cluster 3 (medium happiness group). This indicates that Cluster 1 represents countries with stronger happiness, while Cluster 2 represents countries with lower happiness, and Cluster 3 is at an intermediate level. Second, the per capita GDP shows particularly prominent differences between clusters. The per capita GDP of Cluster 1 is 1.379, which is significantly higher than that of Cluster 3 (1.040) and Cluster 2 (0.537), indicating that the countries in Cluster 1 have stronger economic strength, while the countries in Cluster 2 have relatively weaker economies. Related to this, the difference in social support also indicates the connection between the economy and social security. The social support of Cluster 1 is 1.488, much higher than that of Cluster 2 (0.936) and Cluster 3 (1.337), indicating that the countries in Cluster 1 perform outstandingly in the social security system and mutual assistance network.

The healthy life expectancy shows a similar trend to social support. The healthy life expectancy of Cluster 1 is 0.986, which is significantly higher than that of Cluster 3 (0.821) and Cluster 2 (0.497), showing that the residents in Cluster 1 have a higher health level. The freedom to make life choices is also a significant distinguishing factor between clusters. The

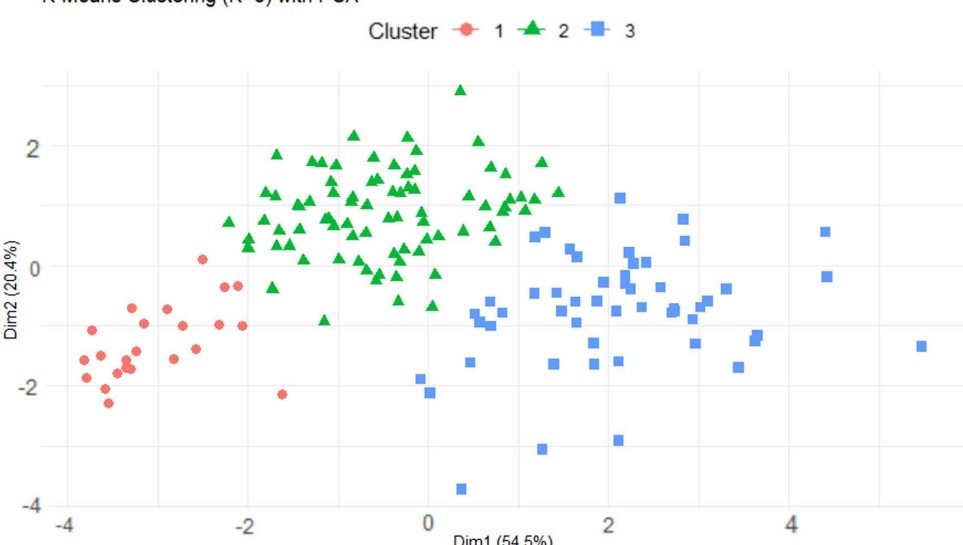

**Fig 6. Clustering Scatter Plot.**

**Table 1. Final Clustering Situation.**

|  | Final Cluster | | |
|---|---|---|---|
|  | 1 | 2 | 3 |
| Score | **7.064** | 4.269 | 5.748 |
| GDP per capita | 1.379 | 0.537 | 1.040 |
| Social support | 1.488 | 0.936 | 1.337 |
| Healthy life expectancy | 0.986 | 0.497 | 0.821 |
| Freedom to make life choices | 0.524 | 0.321 | 0.403 |
| Generosity | 0.245 | 0.190 | 0.157 |
| Perceptions of corruption | 0.241 | 0.094 | 0.074 |

freedom of Cluster 1 is 0.524, which is relatively high, reflecting the advantages of this type of country in civil liberties, while Cluster 3 (0.403) and Cluster 2 (0.321) show relatively low freedom. In general, the countries in Cluster 1 usually have high economic, social support, health levels, and freedom, while the countries in Cluster 2 face lower happiness, economic levels, social support, and healthy life expectancy, and Cluster 3 is in between, showing relatively balanced characteristics.

**3.2.4 Analysis of the correlation matrix after clustering.** As shown in Fig 7, from the perspective of correlation, Score has a strong positive correlation with variables such as GDP per capita, Social support, and Healthy life expectancy. For example, the correlation coefficient between Score and GDP per capita is relatively high and significant, indicating that the economic level is closely related to the happiness score. Social support also has a strong positive correlation with Healthy life expectancy. In terms of the distribution of each variable and clustering, taking GDP per capita as an example, Cluster 1 is more distributed in the area with higher per capita GDP, while Cluster 2 and Cluster 3 are relatively less, showing the economic strength advantage of Cluster 1. Similarly, variables such as Social support also show differences in distribution among different clusters. Cluster 1 is more concentrated

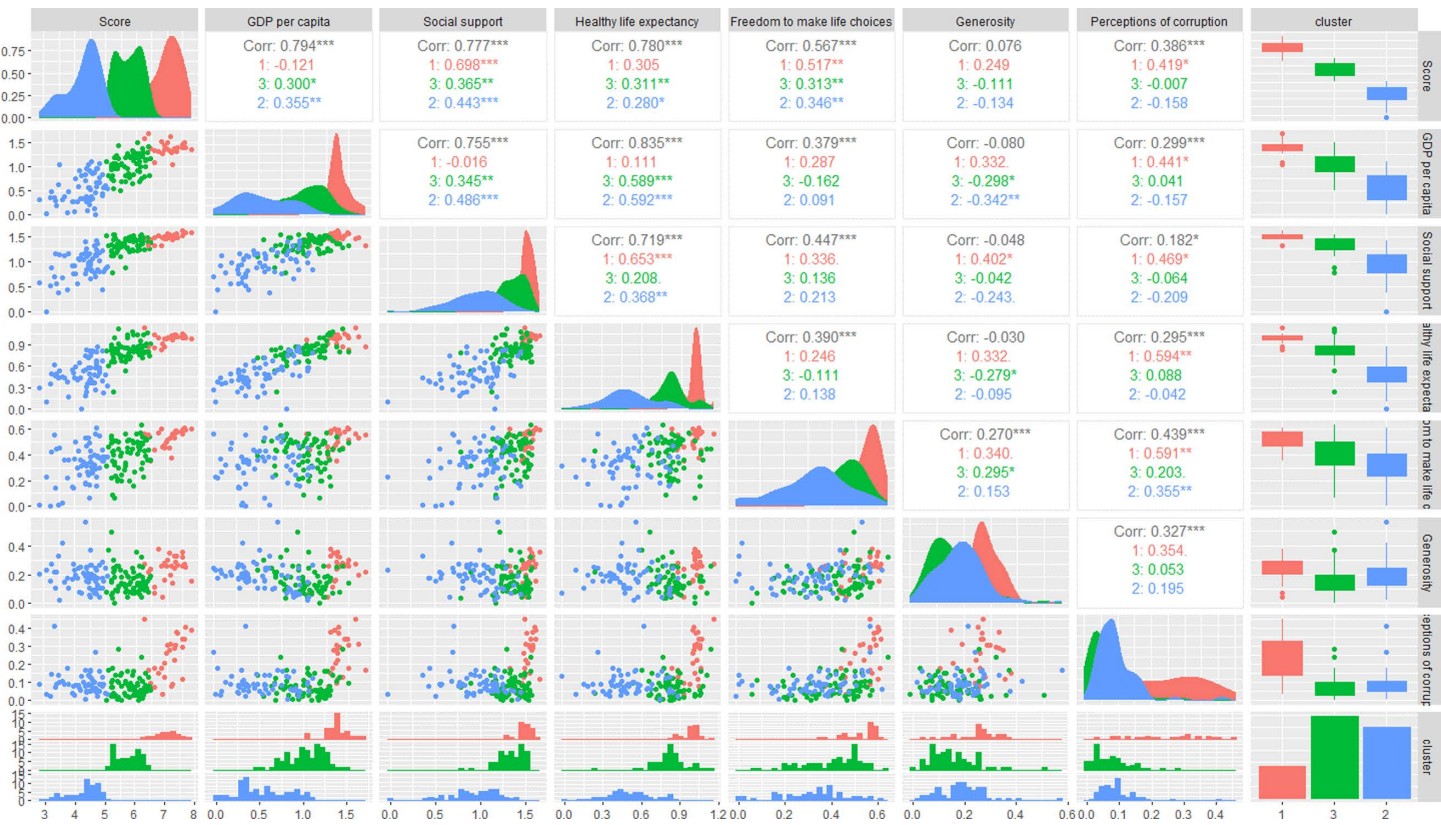

**Fig 7. Correlation Matrix Graph.**

in the high-value area of each variable, while Cluster 2 and Cluster 3 are more dispersed or concentrated in the low-value area. Overall, it clearly shows the correlation between variables and the distribution characteristics of each cluster in different variables, providing more sufficient reference for the machine learning prediction of the study. [29]

## 3.3 Prediction model

**3.3.1 Prediction model building.** After determining the main group characteristics of the happiness index, this study constructs a prediction model based on multiple key variables, mainly using Random Forest and XGBoost algorithms. Through these machine learning models, we can quantitatively evaluate the impact of variables such as GDP, social support, and healthy life expectancy on the happiness index and provide predictions of the future happiness index.

Random Forest is an ensemble learning method based on decision trees. It constructs multiple decision trees and uses the output of each tree to vote to determine the final prediction result [30]. This method has strong predictive ability, can handle nonlinear problems, and avoid overfitting. In this study, Random Forest is used to predict the happiness index, and the grid search method is used to adjust the hyperparameters, such as the number of trees (n_estimators) and the maximum depth (max_depth).

XGBoost is an efficient gradient boosting tree algorithm with strong computing power and robustness, especially suitable for processing large amounts of data and complex nonlinear problems [31]. XGBoost realizes prediction by constructing multiple weak classifiers

and combining them with weights. In this study, XGBoost is used to analyze the influence of factors such as GDP, social support, and healthy life expectancy on the happiness index, and the optimal model parameters are selected through cross-validation.

**3.3.2 Performance evaluation and model comparison.** To evaluate the prediction effect and accuracy of the model, this study uses the following common performance evaluation indicators:

1. Mean square error (MSE): MSE is a commonly used indicator to measure the difference between the predicted value and the actual value. The smaller its value, the higher the prediction accuracy of the model. In this study, MSE is used to evaluate the prediction ability of different models for the happiness index.

2. Coefficient of determination ($R^2$): $R^2$ is an indicator to measure the goodness of fit of the model, with a value range from 0 to 1. The closer it is to 1, the stronger the model's explanatory power for the data. By comparing the $R^2$ values of different models, we can evaluate which model can better explain the changes in the happiness index.

It can be seen from Table 2 that the Random Forest prediction model has better performance. As shown in Fig 8, the fitting situation of the Random Forest prediction model is the best.

## 4 Results

In this study, we conducted a descriptive statistical analysis of the happiness index and its related variables (such as GDP, social support, healthy life expectancy, freedom, generosity, and perceived corruption) of 156 countries. By analyzing the distribution of these variables, we can obtain the characteristics of different countries in the happiness index.

One of the most important results of clustering is that countries in the world can be ranked according to their happiness scores. Since there is a positive correlation between happiness and social development, by knowing the names of the happiest countries in the cluster, it may be possible to guess their social and economic status. Table 3 shows the top 10 countries in each of the 3 clusters ranked by happiness index.

The distribution of key variables in the three groups shows significant differences. High-happiness countries generally have high GDP, social support, healthy life expectancy, and freedom, while low-happiness countries face low economic development levels, social instability, and lack of public services. Through box plot analysis, the distribution of these key variables in different groups can be clearly seen, especially in the social support and GDP variables, the differences between groups are particularly significant.

### 4.1 Hypothesis testing and clustering results

Based on the clustering analysis, we conducted a series of hypothesis tests to verify the significant differences in key variables between different clustering categories [32]. Specifically, we used one-way ANOVA to test whether there are significant differences in variables such as

**Table 2. Performance Evaluation of Machine Learning Prediction Models.**

| *Model* | *MSE* | *R2 Score* |
| --- | --- | --- |
| Linear Regression | 0.393855 | 0.621557 |
| **Random Forest** | **0.219458** | **0.841086** |
| XGBoost | 0.272635 | 0.738033 |

GDP, social support, healthy life expectancy, and freedom between different happiness index groups. The proposed hypotheses are:

*H0*: There are no significant differences in key variables (GDP, social support, healthy life expectancy, etc.) between different happiness index groups.

*H1*: There is at least one variable with a significant difference between different happiness index groups.

Table 4 shows using one-way ANOVA to test variables such as GDP, social support, healthy life expectancy, and freedom. The hypothesis test results support our preliminary hypothesis that countries with higher happiness indexes show significant advantages in multiple key variables. Specifically, GDP, social support, and healthy life expectancy are the key factors affecting the national happiness index, and there is a high positive correlation between these factors.

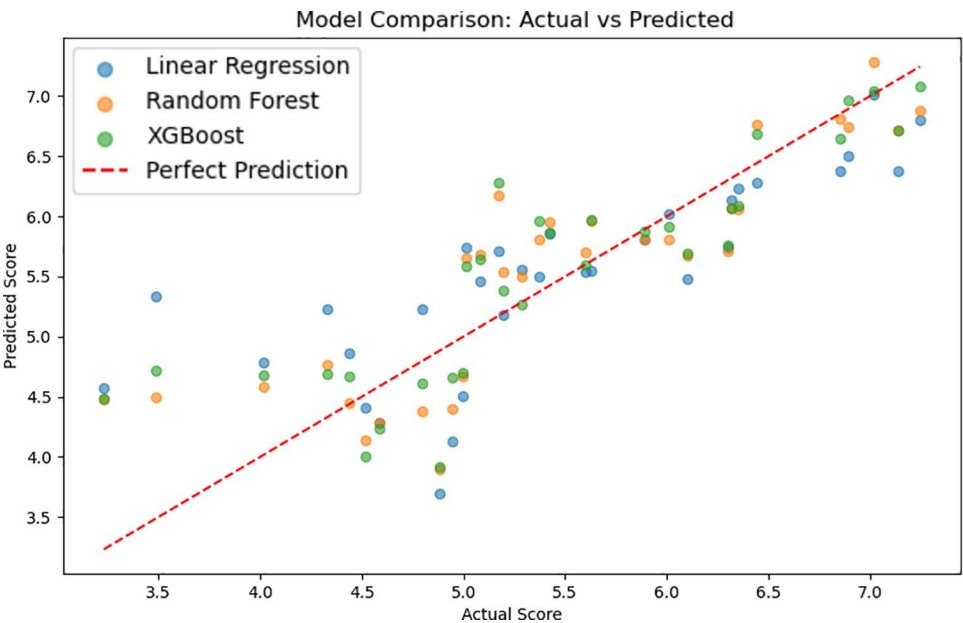

**Fig 8. Fitting Situation of Prediction Model and Actual Value.**

Table 3. Ranking of Different Countries According to Happiness Scores.

| Rank | High happiness | Moderate happiness | Low happiness |
|---|---|---|---|
| 1 | Finland | Chile | Cameroon |
| 2 | Denmark | Guatemala | Ghana |
| 3 | Norway | Saudi Arabia | Ivory Coast |
| 4 | Iceland | Spain | Nepal |
| 5 | Netherlands | Panama | Jordan |
| 6 | Switzerland | Brazil | Benin |
| 7 | Sweden | Uruguay | Congo (Brazzaville) |
| 8 | New Zealand | El Salvador | Gabon |
| 9 | Canada | Italy | Laos |
| 10 | Austria | Bahrain | South Africa |

**Table 4. One-way ANOVA.**

| | Cluster | | Error | | F | Significance |
|---|---|---|---|---|---|---|
| | Mean Square | Degrees of Freedom | Mean Square | Degrees of Freedom | | |
| Score | 79.931 | 2 | 0.210 | 153 | 379.925 | 0.000 |
| GDP per capita | 7.734 | 2 | 0.060 | 153 | 129.562 | 0.000 |
| Social support | 3.861 | 2 | 0.040 | 153 | 96.018 | 0.000 |
| Healthy life expectancy | 2.802 | 2 | 0.023 | 153 | 123.149 | 0.000 |
| Freedom to make life choices | 0.388 | 2 | 0.016 | 153 | 24.701 | 0.000 |
| Generosity | 0.077 | 2 | 0.008 | 153 | 9.372 | 0.000 |
| Perceptions of corruption | 0.283 | 2 | 0.005 | 153 | 52.995 | 0.000 |

Our analysis shows that improving the levels of these key variables, especially GDP and social support, is helpful to improve the national happiness index.

## 5 Discussion

This study systematically investigates the multidimensional characteristics of the global happiness index and its key driving factors through the integration of cluster analysis and machine learning models. The following discussion centers on the three research objectives outlined in Section 1.3, examining their implementation, academic significance, and connections to the existing literature, while further elucidating the study's innovative contributions and practical implications.

Based on multidimensional data from the World Happiness Report, the study employs the K-Means algorithm to classify 156 countries into three groups—high, medium, and low happiness. ANOVA analysis reveals significant differences ($p < 0.001$) among these groups in key variables such as GDP, social support, and life expectancy. For example, the mean social support value in the high-happiness group (1.488) is significantly higher than that in the low-happiness group (0.936), indicating that social support is not only a core variable of the happiness index but also an important indicator of group differences, while revealing its nonlinear role in these differences.

However, the cluster analysis has its limitations. Lower data coverage in some low-income countries (e.g., South Sudan) may affect the representativeness of the groupings. Future research could incorporate field surveys or satellite data (such as the nighttime light index) to supplement economic and social indicators, thereby enhancing the generalizability of the clustering results.

By comparing Random Forest and XGBoost models, this study confirms that the Random Forest model performs optimally in predicting the happiness index. Its variable importance analysis further underscores the central roles of social support and GDP. Traditional linear methods struggle to capture the complex interactions among variables (e.g., the synergistic effects between GDP and social support), whereas machine learning models, with their nonlinear modeling capabilities, more precisely quantify the contribution of multidimensional variables to the happiness index. For instance, the correlation between social support and the happiness index is significantly stronger than that of other variables, suggesting that strengthening the social security network may be a higher policy priority than merely enhancing economic performance.

Based on the clustering and prediction results, the study proposes differentiated policy pathways: high-happiness countries should optimize social support and healthcare protection (e.g., the Nordic welfare model); medium-happiness countries should balance economic growth with social equity (e.g., by increasing public healthcare investment); and

low-happiness countries should prioritize addressing infrastructure and poverty issues (through infrastructure investment). This framework aligns closely with the United Nations Sustainable Development Goals, providing quantitative evidence for policymakers. For example, in medium-happiness countries such as India, the model results indicate that enhancing social support can lead to an increase in the happiness index, thereby offering empirical support for policies like universal healthcare programs.

Nonetheless, the generalizability of these policy recommendations is constrained by the preset selection of variables. The current model does not incorporate cultural or environmental factors (such as religious beliefs or climate change), which may indirectly influence the happiness index by affecting social cohesion. Future research could employ natural language processing techniques to analyze social media texts and extract implicit cultural indicators, thus enhancing the comprehensiveness of the policy recommendations.

The "clustering-prediction" hierarchical framework proposed in this study, which integrates unsupervised and supervised learning for happiness index research, not only improves predictive accuracy but also reveals policy intervention priorities through group difference analysis. This approach can be extended to other social indicators (such as the sustainable development index or regional poverty index), thereby advancing the application of machine learning in the field of public policy.

## 6 Conclusions and perspectives

### 6.1 Theoretical contributions

One of the main contributions of this study is to combine clustering analysis and machine learning, proposing a new method for analyzing the happiness index. This method provides a multi-dimensional analysis framework for happiness index research, filling the gaps in existing research with a single method or variable, especially in modeling the interaction effects of multiple variables, which has important academic value.

Most traditional happiness index studies use regression analysis, correlation analysis, and other methods. These methods mostly rely on assumptions and linear relationships and are difficult to capture the complex nonlinear relationships between variables. In this study, countries are grouped into different clusters according to the happiness index through clustering analysis, and then further analyzed in combination with machine learning models (such as Random Forest and XGBoost), revealing how multiple social and economic factors comprehensively affect the happiness index. This method not only improves the prediction ability of the model but also enhances the understanding of the multidimensional nature of happiness.

In existing research, many studies overemphasize the influence of a single variable (such as GDP or social support) and ignore the interaction between these variables. By comprehensively considering multiple variables such as GDP, social support, and healthy life expectancy, this study proposes a more comprehensive analysis framework for the happiness index, enabling us to consider the effects of multiple variables simultaneously and discover their internal hierarchical structure through clustering analysis. This diversified analysis perspective provides new inspiration for future happiness index research, especially in dealing with complex social and economic data, and can better reveal the interaction relationships between different factors.

### 6.2 Policy suggestions

The practicality of this study lies in providing precise policy basis for governments and policymakers, especially in formulating intervention measures for low and medium-happiness

countries. By deeply analyzing the multidimensional influencing factors of the happiness index, policymakers can more clearly identify the key areas for enhancing happiness and formulate targeted social policies accordingly.

The research shows that the improvement of the happiness index depends not only on a single economic factor but also on the comprehensive effects of social support, health policies, and other aspects. In high-happiness countries, the policy focus should be on further optimizing the social support system, improving the health level of citizens, and promoting social freedom; while in low-happiness countries, the government should give priority to enhancing the level of economic development, improving public health and education services, and strengthening social security and mutual assistance mechanisms.

Low and medium-happiness countries face more challenges in enhancing the happiness of their people. Therefore, different intervention measures need to be taken according to their respective social and economic characteristics. For example, in some low-happiness countries (such as South Sudan and Burundi), improving economic development and infrastructure construction is the top priority for enhancing happiness; while in some medium-happiness countries (such as India and South Africa), the construction of the social security system and the public health system should be strengthened to alleviate poverty and inequality and thus enhance the happiness of the people.

## 6.3 Limitations and mitigations

While this study advances the understanding of global happiness index determinants through novel clustering and predictive modeling, several limitations warrant consideration. Addressing these challenges can guide future research toward more comprehensive and dynamic analyses.

**6.3.1 Data representativeness. Limitations:** The dataset encompasses 156 countries but exhibits significant gaps in low-income regions (e.g., South Sudan, Burundi). Missing values in these areas may compromise the generalizability of clustering results, particularly within the low-happiness subgroup.

**Methodological Response:** Multiple imputation techniques were applied to mitigate bias from incomplete data. [21] However, future studies should prioritize field surveys or alternative data sources (e.g., satellite imagery for economic indicators) to enhance coverage in underrepresented regions.

**6.3.2 Temporal dynamics. Limitations:** The static machine learning models (Random Forest, XGBoost) lack capacity to capture time-varying trends, such as post-pandemic shifts in happiness determinants.

**Methodological Response:** Longitudinal analysis using advanced time-series architectures could enable exploration of how social support, GDP, and other variables interact with external shocks over time.

**6.3.3 Variable selection constraints. Limitations:** The study's variable set, derived from the World Happiness Report, may overlook cultural and environmental factors that indirectly influence well-being.

**Methodological Response:** Expanding variable selection through unstructured data integration (e.g., social media text mining for cultural indicators) and interdisciplinary collaborations with anthropologists/sociologists could enhance model explanatory power.

**6.3.4 Variable selection constraints. Limitations:** The K-Means algorithm's assumption of spherical cluster shapes may inadequately represent non-linear relationships in high-dimensional happiness data.

**Methodological Response:** Robustness checks using alternative clustering methods and hybrid approaches combining dimensionality reduction with clustering are recommended.

By transparently addressing these limitations and proposing actionable solutions, this study lays a foundation for more comprehensive and dynamic analyses of the happiness index, ensuring that future research can build upon its methodological and empirical contributions.

### 6.4 Conclusions

Stratification characteristics of the happiness index: The happiness indexes of countries around the world can be clearly divided into three groups: high-happiness, medium-happiness, and low-happiness. There are significant differences in social and economic characteristics between different groups, especially in key variables such as GDP, social support, and healthy life expectancy, showing significant distribution differences.

Core predictive variables: Social support and GDP are the two core variables affecting the happiness index. High-happiness countries generally have strong social support and high GDP. The advantages of these countries in economic development and social security enable their residents to enjoy higher happiness.

Effectiveness of machine learning methods: Through machine learning algorithms (such as XGBoost and Random Forest), we have successfully constructed a prediction model of the happiness index and verified the importance of social support and GDP in prediction. The XGBoost model performs better than Random Forest and has higher prediction accuracy.

Multidimensional nature of happiness: Happiness is a multidimensional and complex concept. Economic, social, health, cultural, and other factors jointly act on the formation of happiness. Future research should further explore the interaction of these factors and combine longitudinal data and more variables to deepen the understanding of the happiness index.

This research not only reveals the global differences in happiness but also provides a new perspective for the multidimensional analysis of happiness and provides precise policy basis for policymakers, especially for formulating intervention measures for low and medium-happiness countries. Future research can further expand the analysis framework, combine cross-cultural and longitudinal data, and explore the dynamic changes and cross-regional correlations of happiness.

## Supporting Information

**S1 Data. Dataset from the World Happiness Report (2020–2024) covering social and economic indicators of 156 countries and regions.**
(XLSX)

## Acknowledgments

We acknowledge Dr. Xiang Xie for his conceptual guidance and methodological validation.

## Author contributions

**Conceptualization:** Xiang Xie.

**Data curation:** Boxu Yang.

**Formal analysis:** Boxu Yang.

**Methodology:** Boxu Yang.

**Validation of statistical models:** Xiang Xie.

**Writing – original draft:** Boxu Yang.

**Writing – review & editing:** Xiang Xie.

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
