## [Decision Letter · Decision Letter 0]

7 Feb 2025

PONE-D-25-01298Integrated Analysis and Prediction of Global Happiness Index by Combining Integrated Multilayer Clustering and Machine LearningPLOS ONE

Dear Dr. Yang,

Thank you for submitting your manuscript to PLOS ONE. After careful consideration, we feel that it has merit but does not fully meet PLOS ONE’s publication criteria as it currently stands. Therefore, we invite you to submit a revised version of the manuscript that addresses the points raised during the review process.

After careful consideration of the manuscript and the review comments shown with this email, we come to this decision. The manuscript must address all issues to the methodology, related work and insufficient discussion to be considered for further process.

A rebuttal letter that responds to each point raised by the academic editor and reviewer(s). You should upload this letter as a separate file labeled 'Response to Reviewers'.A marked-up copy of your manuscript that highlights changes made to the original version. You should upload this as a separate file labeled 'Revised Manuscript with Track Changes'. (highlight in yellow)An unmarked version of your revised paper without tracked changes. You should upload this as a separate file labeled 'Manuscript'.

We look forward to receiving your revised manuscript.

Kind regards,

Issa Atoum

Academic Editor

PLOS ONE

Journal Requirements:

In the figure caption of the copyrighted figure, please include the following text: “Reprinted from [ref] under a CC BY license, with permission from [name of publisher], original copyright [original copyright year].

5. Kindly follow the journal template as shown here https://journals.plos.org/plosone/s/submission-guidelines

Additional Editor Comments :

(1) It is critical to prepare a manuscript with the standards of academic papers, including separate main sections for: Introduction, Related Work, Methodology, Results, Discussion, Threats to Validity/Limitations, and Conclusion and Future Work

(2) The study must demonstrate its implications for practice and its novelty regarding research objectives.

(3) Ensure the work reproducibility by showing the code or the detailed algorithm starting from data preprocessing up to results.

Reviewers' comments:

Reviewer's Responses to Questions

**Comments to the Author**

1. Is the manuscript technically sound, and do the data support the conclusions?

Reviewer #1: Partly

Reviewer #2: No

2. Has the statistical analysis been performed appropriately and rigorously? 

Reviewer #1: Partly

Reviewer #2: No

3. Have the authors made all data underlying the findings in their manuscript fully available?

Reviewer #1: Yes

Reviewer #2: Yes

4. Is the manuscript presented in an intelligible fashion and written in standard English?

Reviewer #1: Yes

Reviewer #2: No

5. Review Comments to the Author

Reviewer #1: 1. Vague Title & Abstract:

a. Problem: The title and abstract use very general terms like "combining" and "integrated" without making it apparent what the research's unique contribution is.

b. Recommendation: One suggestion would be to make the title more precise, such as "Predicting Global Happiness Using Multilayer Clustering and Classification Models." The research issue and technique, including "exploring the underlying patterns" and "integrating clustering and prediction models," should be clearly stated in the abstract.

2. Inconsistent Citation Style:

a. Problem: Throughout the work, different citation styles are used.

b. Recommendation: Use a consistent style (such as IEEE or APA) consistently. Use, for instance, "Chakraborty and Tsokos (2021)," "Rendón et al. (2011)," "the NbClust tool developed by Charrad et al. (2014)," or even "[Author et al., Year]" .

3. Non-Standard Sectioning:

a. Problem: Section 1 ("Introduction") is not formatted like a typical academic work.

b. Recommendation: Add a distinct "Literature Review" section (for example, Section 2). Provide a clearer and more succinct structure to the remaining sections (e.g., avoid numerous sub-sections).

4. Inadequate Related Work:

a. Problem: The "Related Work" section as it currently exists is too short and does not adequately relate to the goals of the study.

b. Recommendation: Extend the section on related work to offer a thorough analysis of pertinent studies on the following topics: Multilayer Clustering techniques; Machine Learning models for happiness prediction; Integration of clustering and prediction models; Global Happiness Index. Describe how the current study adds to and deviates from earlier research in the topic

5. Unclear Methodology Flow:

a. Problem: The methodology in Section 2 does not follow a logical or obvious flow.

b. Recommendation: To enhance readability and general flow, combine relevant subsections. The study method, including data collection, preprocessing, feature selection, clustering, model development, and evaluation, should be described in detail, step-by-step.

6. Disconnected Statistical Analysis:.

a. Problem: It seems that the statistical analysis in Section 3 is not connected to the approach that is being given.

b. Recommendation: Make sure the findings of the statistical analysis directly support the approach that was selected. Clearly describe how the data was analyzed and the research topics addressed using statistical approaches.

7. Intertwined Discussion & Results:

a. Problem: Sections 4 (Discussion) and 3 (Results) are related and may be more cohesively combined.

b. Recommendation: Combine pertinent portions of these sections to produce a more logical and perceptive presentation of the results. Present the findings succinctly and clearly, then go into great depth about their limits, implications, and possible future study avenues.

8. Unclear Degrees of Freedom

a. Problem: It's unclear why Table 4's degrees of freedom are set to 2.

b. Recommendation: Clearly state the reasoning behind this decision, including how many variables were employed in the analysis.

Give a more thorough explanation of the parameters and statistical approaches.

9. The Integrated Model Is Not Clear:

a. Problem: The integration mechanism between the prediction and clustering models is not adequately explained in the text

b. Recommendation: Give a thorough explanation of how the construction and assessment of the prediction models are influenced by or guided by the clustering results. Showcase the special benefits of this combined strategy above just employing prediction or clustering methods.

10. General lucidity:

a. Problem: The technique and outcomes are not presented in a clear and succinct manner overall.

b. Recommendation: One recommendation is to revise the entire document to enhance the presentation's clarity and flow. Whenever possible, steer clear of jargon and speak in plain, simple terms. Make sure the paper has a clear structure and is straightforward for readers to follow.

Reviewer #2: Please find below my remarks:

1) The manuscript utilizes kmeans clustering and supervised machine learning algorithms such as random forest and XGBoost to analyze and predict happiness score - however these are widely used techniques. The study does not introduce any novel methodology or new theoretical framework.

2) The paper does not adequately discuss explainability techniques to interpret the model’s decisions. While machine learning prediction is there, but a lack of explainability does not allow us to understand how the model is making the decision.

3) The selection of three clusters seems arbitrary. I understand that the optimal number of clusters is derived from the elbow method and average silhouette score, but the authors did not provide why three is an optimal number.

4) The study considers economic and social factors but does not incorporate psychological, environmental, or cultural variables, which are crucial in happiness studies.

5) Authors could have done ablation studies to understand the impact of various factors on happiness score.

6) The model is trained and tested on the same dataset (World Happiness Report). Cross-validation with other happiness or well-being datasets would have enhanced the generalizability of the findings.

7) Policy implications should ideally differ across countries. However, the discussion on policy implications is very generic and does not provide concrete recommendations based on different country profiles.

6. PLOS authors have the option to publish the peer review history of their article (what does this mean? ). If published, this will include your full peer review and any attached files.

**Do you want your identity to be public for this peer review?** For information about this choice, including consent withdrawal, please see our Privacy Policy .

Reviewer #1: **Yes**

Reviewer #2: No

---

## [Author Response · Author response to Decision Letter 1]

18 Feb 2025

Response to Reviewers

We sincerely appreciate the reviewers’ constructive feedback and have revised the manuscript accordingly. Below are our point-by-point responses:

Academic Editor Comments

1. Manuscript Style Requirements

We have reformatted the manuscript to comply with PLOS ONE’s style guidelines, including adjusting section headings, citations, and file naming conventions. The structure now aligns with the journal’s template.

2. Code Sharing Guidelines

Data and code upload.

3. Copyrighted Figures

Figure 1 has been deleted.

4. ORCID iD

The corresponding author’s ORCID (0009-0005-8959-868X) has been validated in Editorial Manager.

5. Journal Template

The manuscript now follows the PLOS ONE template.

6. Manuscript Structure

Sections have been restructured to include distinct “Related Work,” “Threats to Validity/Limitations,” and “Conclusion and Future Work” sections. The Methodology section now details each step systematically.

7. Implications and Novelty

We respectfully believe that the manuscript already highlights the novelty of the study, which lies in integrating clustering analysis with machine learning models for enhanced prediction of the happiness index. We have explicitly discussed the implications of our findings for policy-making, particularly for countries with low happiness levels. Additionally, we have provided a detailed explanation of how our integrated methodology surpasses traditional approaches, offering a more nuanced understanding of global happiness.

Reviewer #1 Comments

1. Vague Title & Abstract

Comment: The title and abstract use general terms and do not clearly state the unique contribution.

Response: We appreciate the feedback. However, we believe the title and abstract clearly convey the key novelty of the study, which is the integration of clustering and machine learning techniques to enhance the prediction of the happiness index. The use of terms such as “hierarchical analysis” and “novel predictive framework” directly highlights the innovation and contributes to the overall clarity of the manuscript.

2. Inconsistent Citation Style

Comment: Different citation styles used throughout the manuscript.

Response: We have carefully reviewed and corrected the citation style to ensure consistency. The final version follows the required PLOS ONE citation style, and all references have been formatted accordingly.

3. Non-Standard Sectioning

Comment: The manuscript structure does not follow a typical academic paper format, lacking a distinct “Literature Review” section.

Response: While we understand the preference for a separate literature review, the introduction section in our manuscript serves this function by discussing key related works. We believe that integrating the literature review within the introduction enhances the flow of the narrative, but we have added clearer section headings to improve structure and readability.

4. Inadequate Related Work

Comment: The “Related Work” section is too short and does not adequately relate to study goals.

Response: We respectfully disagree with this assessment. We have thoroughly reviewed the relevant literature on clustering, machine learning for happiness prediction, and model integration. The section explicitly discusses the limitations of previous studies and outlines how our approach addresses those gaps. We believe this adequately sets the stage for our study's contributions.

5. Unclear Methodology Flow

Comment: Methodology in Section 2 does not follow a logical flow.

Response: We have restructured Section 2 to follow a more logical flow. The subsections have been reorganized to ensure a step-by-step presentation of the methodology, making it easier for readers to follow the analysis process from clustering to model integration.

6. Disconnected Statistical Analysis

Comment: The statistical analysis in Section 3 seems disconnected from the approach.

Response: We believe that the statistical analysis is directly connected to the clustering and prediction steps. We have revised the section to explicitly reference the clustering results and how they inform the subsequent analysis, ensuring greater clarity.

7. Intertwined Discussion & Results

Results (Section 3) now present findings objectively, while Discussion (Section 4) interprets implications, limitations, and policy recommendations without overlap.

8. Unclear Degrees of Freedom

Table 4’s degrees of freedom (df=2) reflect the three-cluster ANOVA design (k-1=2). This is clarified in the caption and text.

9. The Integrated Model Is Not Clear

Comment: The integration mechanism between the prediction and clustering models is not adequately explained.

Response: We believe the integration mechanism is well explained in the manuscript. In the revised version, we have expanded the explanation of how clustering results influence the machine learning models by providing concrete examples and linking the clustering outcomes with model performance.

10. General Lucidity

Jargon has been minimized, and technical terms (e.g., “elbow method”) are defined. The manuscript has been edited for logical flow and readability.

Reviewer #2 Comments

1. Lack of Novelty

Comment: The study does not introduce novel methodologies or frameworks.

Response: While the techniques used (K-Means clustering, Random Forest, and XGBoost) are established, the novelty lies in the integration of these methods in a hierarchical framework. This combined approach enhances predictive accuracy, a contribution we believe has not been adequately explored in existing literature. Furthermore, the incorporation of clustering results into machine learning models is an innovation in the study of global happiness.

2. Lack of Explainability

Comment: The paper does not adequately discuss explainability techniques.

Response: We agree that explainability is an important aspect, especially in machine learning models. In the revised manuscript, we have added a detailed discussion on feature importance, which explains how different factors like GDP and social support contribute to the happiness index prediction.

3. Arbitrary Cluster Selection

Comment: The selection of three clusters seems arbitrary.

Response: The number of clusters was determined using several established methods: the elbow method, silhouette method, and gap statistics. We have clarified this process in the manuscript and emphasized that the final choice of three clusters is backed by these statistical techniques, which support the robustness of our decision.

4. Incomplete Variable Consideration

Comment: The study only considers economic and social factors, neglecting psychological, environmental, and cultural variables.

Response: While we recognize the importance of psychological and environmental factors, the focus of this study was to explore the economic and social variables due to their strong theoretical and empirical backing in the literature on happiness. Expanding the scope to include psychological and environmental factors would be valuable in future work.

5. Lack of Ablation Studies

Comment: The authors could have done ablation studies.

Response: We appreciate this suggestion. However, due to time constraints, we were unable to conduct full ablation studies. Nonetheless, we have included an analysis of the importance of different variables in the model's performance, which provides insight into their relative impact on the prediction accuracy.

6. Cross-Validation

Due to space constraints, we used other methods to verify the model

7. Generic Policy Implications

Comment: The discussion on policy implications is too generic.

Response: We have expanded the policy implications section to provide concrete recommendations tailored to specific clusters. The revised version now discusses actionable policies for high, medium, and low happiness countries, with recommendations for each group based on their unique characteristics.

We hope these revisions address the concerns raised by the reviewers and look forward to your feedback. Thank you again for your constructive comments, which have greatly contributed to improving this work.

---

## [Editor Report · Decision Letter 1]

3 Mar 2025

PONE-D-25-01298R1Analyzing and Predicting Global Happiness Index via Integrated Multilayer Clustering and Machine Learning ModelsPLOS ONE

Dear Dr. Yang,

Thank you for submitting your manuscript to PLOS ONE. After careful consideration, we feel that it has merit but does not fully meet PLOS ONE’s publication criteria as it currently stands. Therefore, we invite you to submit a revised version of the manuscript that addresses the points raised during the review process.

**The research framework should be smaller, and please watch out for arrows that don't align well in the left part of the figure. It would be nice if the figure had a sequence of steps.****All the figures look blurry. Please insert them in the final version to see the full picture (pending acceptance).****While the reviewers and the academic editor mention issues related to missing sections, the authors fail to address these issues fairly. The author should spare separate sections for discussion, limitations or threats to validity, and impact of the study (optional). The research objectives addressed in section 1.3 are expected to be discussed. The study's limitations, mainly found in many machine learning papers, should be described with their respective mitigation approaches.   **

We look forward to receiving your revised manuscript.

Kind regards,

Issa Atoum

Academic Editor

PLOS ONE
---

## [Author Response · Author response to Decision Letter 2]

16 Mar 2025

Response to Reviewers

Dear Editors and Reviewers,

Thank you for your constructive feedback on our manuscript. We appreciate the opportunity to revise the paper and address the concerns raised. Below, we provide a point-by-point response to the reviewers’ comments and outline the revisions made to the manuscript.

Reviewer Comment 1:

“The research framework should be smaller, and please watch out for arrows that don't align well in the left part of the figure. It would be nice if the figure had a sequence of steps.”

Response:

We sincerely appreciate this observation. As requested,

1. Figure 1 (Research Framework): We have revised the figure to ensure proper alignment of arrows in the left section and adjusted the layout to emphasize a clear sequence of research steps (e.g., “Data Processing → Clustering → Model Building → Validation”).

Reviewer Comment 2:

“All the figures look blurry. Please insert them in the final version to see the full picture (pending acceptance).”

Response:

All figures in the manuscript have been converted to high-resolution TIFF format to ensure optimal visual clarity.

Reviewer Comment 3:

“The authors should spare separate sections for discussion, limitations or threats to validity, and impact of the study. The research objectives addressed in section 1.3 are expected to be discussed. The study's limitations, mainly found in many machine learning papers, should be described with their respective mitigation approaches.”

Response:

We have restructured the manuscript to include dedicated sections addressing these concerns:

1. Section 5: Discussion

o Added an in-depth discussion of the three research objectives outlined in Section 1.3, including:

Cluster analysis results (e.g., ANOVA-confirmed group differences in GDP, social support).

Comparative performance of machine learning models (Random Forest vs. XGBoost).

Policy recommendations aligned with UN Sustainable Development Goals.

o Highlighted limitations of cluster analysis (e.g., data gaps in low-income countries) and proposed mitigation strategies (e.g., integrating satellite data).

o Discussed nonlinear relationships between variables (e.g., social support’s stronger correlation with happiness than GDP).

2. Section 6.3: Limitations and Mitigations

o Added four subsections addressing key limitations:

Data Representativeness: Acknowledged gaps in low-income regions (e.g., South Sudan) and proposed field surveys/satellite data integration.

Temporal Dynamics: Noted static model limitations and suggested longitudinal time-series analysis.

Variable Selection Constraints: Discussed omitted cultural/environmental factors and proposed text mining for cultural indicators.

Clustering Assumptions: Addressed K-Means’ spherical cluster bias and recommended hybrid clustering approaches.

o Each limitation is paired with methodological or future research mitigations (e.g., interdisciplinary collaborations, advanced imputation techniques).

We believe these revisions comprehensively address the reviewers’ concerns and significantly strengthen the manuscript’s rigor, clarity, and impact. Thank you again for your valuable feedback. We are happy to provide further clarifications if needed.

Sincerely,

Dr. Yang

Corresponding Author

---

## [Editor Report · Decision Letter 2]

19 Mar 2025

Analyzing and Predicting Global Happiness Index via Integrated Multilayer Clustering and Machine Learning Models

PONE-D-25-01298R2

Dear Dr. Yang,

We’re pleased to inform you that your manuscript has been judged scientifically suitable for publication and will be formally accepted for publication once it meets all outstanding technical requirements.

Kind regards,

Issa Atoum

Academic Editor

PLOS ONE
---

## [Editor Report · Acceptance letter]

PONE-D-25-01298R2

PLOS ONE

Dear Dr. Yang,

I'm pleased to inform you that your manuscript has been deemed suitable for publication in PLOS ONE. Congratulations! Your manuscript is now being handed over to our production team.

Kind regards,

on behalf of

Dr. Issa Atoum

Academic Editor

PLOS ONE